**Data Availability Statement:** The Institutional Review Board of the "National Institute of Gastroenterology "S. De Bellis" (at Istituto Tumori

# Cross-sectional relationship among different anthropometric parameters and cardio-metabolic risk factors in a cohort of patients with overweight or obesity

**Luisa Lampignano**[1], **Roberta Zupo**[1], **Rossella Donghia**[1], **Vito Guerra**[1], **Fabio Castellana**[1], **Isanna Murro**[2], **Carmen Di Noia**[2], **Rodolfo Sardone**[1], **Gianluigi Giannelli**[3], **Giovanni De Pergola**[2]*

1 Population Health Unit—"Salus in Apulia Study" National Institute of Gastroenterology "Saverio de Bellis", Research Hospital, Castellana Grotte, Bari, Italy, 2 Department of Biomedical Science and Human Oncology, University of Bari, School of Medicine, Policlinico, Bari, Italy, 3 Scientific Direction, National Institute of Gastroenterology "Saverio de Bellis", Research Hospital, Castellana Grotte, Bari, Italy

* gdepergola@libero.it

## Abstract

### Background

Body fat distribution influences the risk of cardio-metabolic disease in people with over-weight. This study was aimed at identifying the anthropometric parameters more strongly associated with the majority of cardio-metabolic risk factors.

### Methods

This study included 1214 subjects (840 women), with a body-mass-index (BMI) $\geq$ 25 Kg/m2, aged 39.2 ± 13 years. Fasting blood glucose (FBG), triglycerides (TG), total, HDL- and LDL-cholesterol, uric acid, vitamin D, high-sensitive C-reactive protein (hs-CRP), white blood cells (WBC), platelets, insulin and insulin resistance (HOMA-IR), systolic (SBP) and diastolic blood pressure (DBP), smoking habit and snoring were evaluated as cardio-metabolic risk factors.We also included the Systematic COronary Risk Evaluation (SCORE) to estimate cardiovascular risk in our study population. BMI, waist circumference (WC), waist-to-height-ratio (WHtR) and neck circumference (NC) were evaluated as anthropometric parameters.

### Results

All four anthropometric parameters were positively associated to SBP, DBP, TG, FBG, insulin, HOMA-IR, WBC, and snoring (p<0.001), and negatively associated with HDL-cholesterol (p<0.001). NC showed a positive association with LDL-cholesterol (β = 0.76; p = 0.01; 95% C.I. 0.19 to 1.32), while vitamin D was negatively associated to WC (β = -0.16; p<0.001; 95% C.I. -0.24 to -0.09), BMI (β = 0.42); p<0.001; 95% C.I. -0.56 to -0.28) and WHtR (β = -24.46; p<0.001; 95% C.I. -37 to -11.9). Hs-CRP was positively correlated with WC (β = 0.003; p = 0.003; 95% C.I. 0.001 to 0.006), BMI (β = 0.01; p = 0.02; 95% C.I. 0.001

"Giovanni Paolo II" I.R.C.C.S) states that it contains potentially identifying and sensitive patient information. For this reason, any request may be addressed directly to the IRB comitatoetico@oncologico.bari.it.

**Funding:** The authors received no specific funding for this work.

**Competing interests:** The authors have declared that no competing interests exist.

to 0.012) and WHtR (β = 0.55; p = 0.01; 95% C.I. 0.14 to 0.96). SCORE was associated to NC (*β = 0.15*; 95% CI 0.12 to 0.18; p<0.001), BMI (*β = -0.18*; 95% CI -0.22 to 0.14; p<0.001) and WHtR (*β = 7.56*; 95% CI 5.30 to 9.82; p<0.001).

## Conclusions

NC, combined with BMI and WC or WHtR could represent an essential tool for use in clinical practice to define the cardio-metabolic risk in individuals with excess body weight.

## Introduction

The obesity epidemic is recognized as one of the most important public health problems in the world today; in most European countries, the prevalence of overweight and obesity exceeds 60% [1]. Obesity has been defined as a risk factor for several cardiovascular (CV) risk factors, including hypertension, type II diabetes, and dyslipidemia [2], and shown to be responsible for higher morbidity and mortality rates in cardiovascular disease (CVD) [3]. Accordingly, a recent systematic review and meta-analysis examining 95 cohorts showed that obesity was associated with a nearly 60% higher prevalence of CVD, as compared to normal weight figures [4]. While U.S. CVD mortality rates have declined overall in the past decades, the rate of decline has recently decelerated, possibly due to the obesity epidemic, contributing to reverse the CVD progress previously obtained [5]. Several studies showed that a subgroup of subjects with obesity may be at significantly lower risk than usually estimated from obesity-related CVDs [6]. This subset has been described as Metabolically healthy obesity (MHO) [6]. Compared to patients with metabolically unhealthy obesity, individuals with MHO are distinguished by lower liver and visceral fat but higher subcutaneous leg fat content, higher cardiorespiratory fitness and physical activity, insulin sensitivity, lower levels of inflammatory markers, and normal adipose tissue function [6].

BMI is commonly used to define the diagnosis of obesity but alone, it is not sufficient to properly assess or manage the cardio-metabolic risk associated with increased adiposity in adults [7,8]. In fact, it is well known that body fat distribution (BFD) is more important than BMI in defining the CVD risk. In particular, despite decades of unequivocal evidence that waist circumference (WC) provides additional, independent information to BMI in predicting morbidity and risk of death, only recently was a suitable Consensus Statement proposed, advising routine measurement of WC by practitioners, as an important opportunity to improve patients management and health [8]. The same Consensus recommended that a decrease in waist circumference, and not in body weight or in BMI, is the most important treatment target, reducing adverse health risks in both men and women [8].

Interestingly, additional anthropometric measurements have been implemented to describe BFD and examine the relationship between BFD and CVD risk. It would be worthwhile verifying whether these parameters are more informative than WC in predicting the cardio-metabolic risk. Excluding methods requiring specific instruments (ultrasounds CT, MRI), the waist to hip ratio (WHR), waist to height ratio (WHtR) and neck circumference (NC) are alternative methods to WC which can be applied to examine anthropometric factors. However, WHR has been progressively abandoned since WC has been demonstrated to be more accurate in quantifying the CVD risk in patients with obesity [9].

NC is commonly utilized as an anthropometric marker to detect patients at higher risk of developing the obstructive sleep apnea syndrome (OSAS) [10], whereas few studies have used it to identify patients affected by metabolic disorders [11]. Recently, WHtR was suggested as a

simpler indicator of abdominal obesity, with greater practical advantages than BMI and WC [12]. Moreover, some reviews highlighted the superiority of WHtR in predicting cardio-metabolic risks among adults and adolescents, while its interpretation can be applied to different ethnic groups and does not require sex-dependent or age-dependent cut-offs [13,14]. Despite this, other reviews showed no differences in predictive powers for CVD risk factors among the various anthropometric indices [15,16].

Only one study (the SOON cohort) has yet compared anthropometric parameters with the aim of identifying which is the best cardio-metabolic risk marker in subjects with obesity, and it showed that neck circumference was the most appropriate anthropometric marker [15]. However, this study investigated only women and only patients with severe obesity and, therefore, a population that is not representative of the whole population with overweight and obesity [14]. Moreover, it did not investigate WHtR [17].

To the best of our knowledge, no study has ever compared BMI, WC, WHtR and NC in relation to CV risk factors in a wide population of men and women affected by excess body weight. For this reason, the present study was focused on exploring the cross-sectional association among the most commonly used anthropometric parameters, namely BMI, WC, WHtR and NC and the CV risk factors mainly evaluated in clinical practice, such as systolic and diastolic blood pressure, and fasting glucose, lipid (triglycerides, total, HDL and LDL cholesterol), insulin, $HOMA_{IR}$ [18], uric acid, 25-hydroxyvitamin D (25(OH)D) [19], C-reactive protein (CRP), white blood cells and platelets numbers in a population of 1214 apparently healthy subjects with overweight and obesity. We further explored the association of the anthropometric parameters using the Systematic COronary Risk Evaluation (SCORE), consisting in charts evaluating the cardiovascular risk by gender, age, total cholesterol, systolic blood pressure and smoking status, provided by the European Society of Cardiology [20].

## Materials and methods

### Study population and design

This cross-sectional study included 1214 consecutive patients (840 females and 374 males, aged 39.2 ± 13 years, all Caucasian) enrolled from January 2018 to December 2019 at the Outpatients Clinic of Nutrition of the Medical Oncology Unit, Department of Biomedical Sciences and Human Oncology, University of Bari, School of Medicine, Policlinico, Bari, Italy and at the "Population Health Unit" of the National Institute of Gastroenterology "S. de Bellis," Research Hospital, Castellana Grotte, Apulia, Italy. Inclusion criteria were a condition of overweight or obesity (BMI ≥ 25 Kg/m2), and taking no medication, including oral contraceptives or drugs for osteoporosis. Exclusion criteria were any history of endocrinological diseases (diabetes mellitus, hypo or hyperthyroidism, hypopituitarism, etc.), chronic inflammatory diseases, stable hypertension, angina pectoris, stroke, transient ischemic attack, heart infarction, congenital heart disease, any malignancies, renal and liver failure, inherited thrombocytopenia.

Prior approval by the Institutional Review Board of the "National Institute of Gastroenterology "S. De Bellis" of this study protocol (Clinical Trial NCT04318288), with its measurements and data collections, was obtained in accordance with the 1964 Helsinki Declaration and subsequent revisions. All participants provided written informed consent to enter the study. The study adhered to the "Strengthening the Reporting of Observational Studies in Epidemiology" (STROBE) guidelines (https://www.strobe-statement.org/).

### Clinical, anthropometric and biochemical parameters assessment

All subjects were closely examined for medical history, hormonal, metabolic and routine hematochemical parameters. Extemporaneous ambulatory diastolic (DBP) and systolic blood

pressure (SBP) was determined in a sitting position after at least a 10-min rest, three different times, using an OMRON M6 automatic Blood Pressure monitor. The final values of blood pressure (SBP and DBP) were the mean of the last two of three measurements.

All anthropometric measurements were taken with participants wearing lightweight clothing and no shoes. All variables were collected at the same time between 8:00 and 10:00 a.m., following an overnight fast. Height was measured to the nearest 0.5 cm using a wall-mounted stadiometer (Seca 711; Seca, Hamburg, Germany). Body weight was determined to the nearest 0.1 kg using a calibrated balance beam scale (Seca 711; Seca, Hamburg, Germany). BMI was calculated by dividing body weight (Kg) by the square of height (m$^2$) and classified according to World Health Organization criteria for normal weight (18.5–24.9 kg/m2), overweight (25.0–29.9 kg/m2), grade I obesity (30.0–34.9 kg/m2), grade II obesity (35.0–39.9 kg/m2), and grade III obesity ($\geq$40.0 kg/m2) [21]. Waist circumference (WC) was measured at the narrowest part of the abdomen, or in the area between the tenth rib and the iliac crest (minimum circumference). Neck circumference (NC) was measured below the laryngeal prominence and perpendicular to the long axis of the neck, and the minimal circumference was recorded to the nearest 0.1 cm [22]. Both circumferences were measured with a Seca 201 ergonomic circumference measuring tape. Waist to height ratio WHtR was calculated by dividing WC (cm) by height (cm) [23].

Blood samples were drawn between 08:00 h and 09:00 h after overnight fasting. Blood glucose (FBG), insulin, 25(OH)D, total cholesterol, high- and low-density lipoprotein (HDL, LDL) cholesterol, triglycerides, Red Blood Cells (RBC), White Blood Cells (WBC), platelets, uric acid, and high sensitive C-reactive protein (hsCRP) serum levels were assayed. Serum insulin concentrations were measured by radioimmunoassay (Behring, Scoppito, Italy). Serum 25(OH)D levels were quantified by chemiluminescence (Diasorin Inc., Stillwater, OK, USA) and all samples were analyzed in duplicate. Plasma glucose was determined using the glucose oxidase method (Sclavus, Siena, Italy), while the concentrations of plasma lipids (triglycerides, total cholesterol, HDL cholesterol) were quantified by automated colorimetric method (Hitachi; Boehringer Mannheim, Mannheim, Germany). LDL cholesterol was calculated by applying the Friedewald equation. Serum uric acid was measured by the URICASE/POD method implemented on an autoanalyzer (Boehringer Mannheim, Mannheim, Germany). Blood cell count was determined using an XT-2000i hematology analyzer (Sysmex, Dasit, Cornaredo, Italy). Hs-CRP was measured on a Cobas Integra 400 Plus using a latex particle-enhanced immunoturbidimetric assay following the manufacturer's instructions (Roche Diagnostics, Indianapolis, IN) [24]. Insulin resistance was assessed with the Homeostasis Model Assessment–Insulin Resistance (HOMA-IR) [25]. Smoking status was assessed on the single question "Are you a current smoker?", categorized as yes or no. Snoring status was assessed by asking the partner or the patient under observation the question "do you usually snore?", categorized as yes or no. Metabolic Syndrome was assessed with International Diabetes Federation (IDF) criteria [26] MHO was identified in people who had BMI of over 30, but they do not have Metabolic Syndrome [6].

## Statistics

Mean and standard deviation (M±SD) for continuous variables, and frequency for categorical variables were used, as indices of centrality. Linear and logistic regression models, adjusted for age and sex, were used to evaluate the association between single variables and different anthropometric parameters. We applied Poisson regression to test the association with the ordinal dependent variable (SCORE). To explore the association between all anthropometric parameters and the risk of cardiovascular events, we used a stepwise regression method,

applying backward selection of associated variables. This method also allowed us to estimate the association of all those variables independently from the mutual collinearity. For those patients below the age limit (40 years) we attributed a risk of "0" for this calculator. When testing the null hypothesis, the probability level of α error was 0.05, two-tailed. All statistical analyses were performed using STATA 16, StataCorp. 2019. Stata Statistical Software: Release 16. College Station, TX: StataCorp LLC.

## Results and discussion

Table 1 shows the general, anthropometric, hormone, metabolic and routine biochemical characteristics of the enrolled population. Our study population consisted of 31% men, mean age was about 40 years (range 14–70 years) and mean BMI was 34 kg/m$^2$ (range 25–64.4).

Table 2 shows the association between each single anthropometric parameter with the different variables investigated in the study, evaluated by linear and/or logistic regression models after adjustment for age and gender. All four anthropometric parameters were positively associated to DBP, SBP, triglycerides, FBG, insulin, HOMA-IR, WBC, and negatively associated with HDL-cholesterol (p<0.001). Only NC showed a positive association with LDL-cholesterol (p = 0.01), while Vitamin D was negatively associated to WC (p<0.001), BMI (p<0.001) and WHtR (p<0.001), but not to NC. Lastly, CRP was positively correlated with WC (p = 0.003), BMI (p = 0.02) and WHtR (p = 0.01), but not with NC. At logistic regression, all the

**Table 1. Characteristics of the study population (n = 1214).**

| Parameters* | | Range (min–max) |
|---|---|---|
| Sex (M) (%) | 375 (30.89) | – |
| Age (years) | 39.50±12.61 | 14.00–74.00 |
| Smoking (Yes) (%) | 246 (20.26) | – |
| Snoring (Yes) (%) | 649 (53.46) | - |
| Neck (cm) | 41.01±4.01 | 30.00–57.00 |
| BMI (Kg/m$^2$) | 33.89±6.08 | 25.00–64.60 |
| Waist (cm) | 108.57±14.09 | 69.00–158.00 |
| Waist to Height Ratio | 0.66±0.08 | 0.25–1.12 |
| DBP (mmHg) | 81.33±9.85 | 55.00–120.00 |
| SBP (mmHg) | 125.76±14.41 | 90.00–180.00 |
| Total Cholesterol (mg/dL) | 192.49±38.89 | 51.00–372.00 |
| Triglyceride (mg/dL) | 107.49±61.49 | 23.00–541.00 |
| FBG (mg/dL) | 91.90±12.55 | 65.00–125.00 |
| HDL Cholesterol (mg/dL) | 48.35±12.47 | 19.00–116.00 |
| Insulin (mg/dL) | 22.82±16.18 | 2.40–128.00 |
| HOMA-IR | 5.29±4.11 | 0.52–34.70 |
| LDL Cholesterol (mg/dL) | 123.40±33.75 | 23.00–262.00 |
| Platelets (10$^3$/μL) | 264.49±59.43 | 249–368.00 |
| WBC (10$^3$/μL) | 7.05±1.63 | 3.26–13.44 |
| Vitamin D (ng/dL) | 19.16±5.81 | 4.00–50.40 |
| hs-CRP (mg/dL) | 0.45±0.45 | 0.00–6.30 |

* Mean and standard deviation (M±SD) for continuous variables. Percentage (%) for categorical variables.
Abbreviations: BMI, Body Mass Index; DBP, Diastolic Blood Pressure; SBP, Systolic Blood Pressure; FBG, Fasting Bllod Glucose; HDL, Hight Density Lipoprotein; HOMA-IR, Homeostasis Model Assessment-Insulin Resistance; LDL, Low Density Lipoprotein; WBC, White Blood Cell; hs-CRP, high sensitive C-Reactive Protein.

**Table 2. Multivariate regression models # between continuous § and categorical ψ variables and anthropometric parameters.**

| Parameters * | NC (cm) | | | BMI (Kg/m²) | | | WC (cm) | | | WHtR | | |
|---|---|---|---|---|---|---|---|---|---|---|---|---|
| | β | p-value | C.I. (95%) | β | p-value | C.I. (95%) | B | p-value | C.I. (95%) | β | p-value | C.I. (95%) |
| *Continuous Variables §* | | | | | | | | | | | | |
| DBP (mmHg) | **0.40** | **<0.001** | **0.23 to 0.57** | **0.16** | **<0.001** | **0.07 to 0.24** | **0.09** | **<0.001** | **0.05 to 0.13** | **11.96** | **<0.001** | **5.98 to 17.94** |
| SBP (mmHg) | **0.58** | **<0.001** | **0.33 to 0.84** | **0.35** | **<0.001** | **0.23 to 0.47** | **0.16** | **<0.001** | **0.11 to 0.22** | **22.06** | **<0.001** | **13.43 to 30.70** |
| Total Cholesterol (mg/dL) | 0.49 | 0.14 | -0.16 to 1.14 | 0.02 | 0.89 | -0.30 to 0.35 | 0.04 | 0.59 | -0.11 to 0.19 | 9.89 | 0.41 | -13.45 to 33.23 |
| Triglyceride (mg/dL) | **3.67** | **<0.001** | **2.64 to 4.70** | **2.04** | **<0.001** | **1.52 to 2.56** | **0.88** | **<0.001** | **0.65 to 1.11** | **132.64** | **<0.001** | 95.48 to 169.79 |
| FBG (mg/dL) | **0.58** | **<0.001** | **0.37 to 0.78** | **0.32** | **<0.001** | **0.21 to 0.42** | **0.14** | **<0.001** | **0.09 to 0.19** | **19.74** | **<0.001** | **12.13 to 27.35** |
| HDL (mg/dL) | **-0.92** | **<0.001** | **-1.14 to -0.71** | **-0.42** | **<0.001** | **-0.53 to -0.32** | **-0.19** | **<0.001** | **-0.24 to -0.15** | **-28.84** | **<0.001** | **-36.16 to -21.53** |
| Insulin (mg/dL) | **1.80** | **<0.001** | **1.54 to 2.06** | **1.10** | **<0.001** | **0.97 to 1.24** | **0.45** | **<0.001** | **0.39 to 0.51** | **65.75** | **<0.001** | **56.00 to 75.51** |
| HOMA-IR | **0.46** | **<0.001** | **0.39 to 0.53** | **0.27** | **<0.001** | **0.24 to 0.31** | **0.11** | **<0.001** | **0.10 to 0.13** | **16.11** | **<0.001** | **13.60 to 18.62** |
| LDL Cholesterol (mg/dL) | **0.76** | **0.01** | **0.19 to 1.32** | 0.07 | 0.63 | -0.22 to 0.37 | 0.07 | 0.25 | -0.05 to 0.20 | 13.44 | 0.20 | -7.33 to 34.22 |
| Platelets (10³/μL) | 1.11 | 0.09 | -0.17 to 2.39 | **0.96** | **0.001** | **0.39 to 1.54** | **0.40** | **0.002** | **0.14 to 0.65** | **56.54** | **0.007** | **15.74 to 97.33** |
| WBC (10³/μL) | **0.04** | **0.02** | **0.01 to 0.08** | **0.02** | **0.01** | **0.01 to 0.04** | **0.01** | **0.004** | **0.003 to 0.019** | **1.41** | **0.02** | **0.21 to 2.60** |
| Vitamin D (ng/dL) | -0.32 | 0.12 | -0.74 to 0.09 | **-0.42** | **<0.001** | **-0.56 to -0.28** | **-0.16** | **<0.001** | **-0.24 to -0.09** | **-24.46** | **0.001** | **-37.00 to -11.93** |
| hs-CRP (mg/dL) | 0.005 | 0.27 | -0.004 to 0.014 | **0.01** | **0.02** | **0.001 to 0.012** | **0.003** | **0.003** | **0.001 to 0.006** | **0.55** | **0.01** | **0.14 to 0.96** |

| Parameters * | Neck (cm) | | | BMI (Kg/m²) | | | Waist (cm) | | | Waist to Height Ratio | | |
|---|---|---|---|---|---|---|---|---|---|---|---|---|
| | OR | p-value | C.I. (95%) | OR | p-value | C.I. (95%) | OR | p-value | C.I. (95%) | OR | p-value | C.I. (95%) |
| *Dichotomic Variables ψ* | | | | | | | | | | | | |
| Snoring | **1.16** | **<0.001** | **1.11 to 1.21** | **1.09** | **<0.001** | **1.06 to 1.11** | **1.04** | **<0.001** | **1.03 to 1.05** | **217.23** | **<0.001** | **43.36 to 1088.34** |
| Smoking | 1.01 | 0.66 | 0.96 to 1.06 | 1.01 | 0.56 | 0.98 to 1.03 | 1.00 | 0.34 | 0.99 to 1.01 | 1.96 | 0.40 | 0.41 to 9.31 |

§ Multivariate Linear Regression Model

ψ Multivariate Logistic Regression Model.

# Adjusted for Age and Gender.

* Abbreviations: β, coefficient; C.I., Coefficient Interval at 95%; OR, Odds Ratios; NC, neck circumference; BMI, Body Mass Index; WC, waist circumference; WHtR, waist to height ratio; DBP, Diastolic Blood Pressure; SBP, Systolic Blood Pressure; FBG, Fasting Blood Glucose; HDL, HighDensity Lipoprotein; HOMA-IR, Homeostasis Model Assessment-Insulin Resistance; LDL, Low Density Lipoprotein; WBC, White Blood Cell; hs-CRP, high sensitive C-Reactive Protein.

anthropometric parameters were positively associated with snoring (p<0.0001). No further associations were evident in the models.

Table 3 shows the association between the SCORE and all anthropometric parameters evaluated (A), also applying a backward stepwise method for SCORE on all variables included together in the model (B). In the final model the SCORE was significantly associated with NC (*β = 0.15*; 95% CI 0.12 to 0.18; p<0.001), followed by BMI (*β = -0.18*; 95% CI -0.22 to 0.14; p<0.001) and WHtR (*β = 7.56*; 95% CI 5.30 to 9.82; p<0.001). We also performed the same models in two different subgroups of our study population: subjects with Metabolic Syndrome in (representing 36.9% of the total group) and people with MHO (48.3%). In the first subgroup (Table 4) SCORE was associated to NC (*β = 0.06*; 95% CI 0.01 to 0.10; p = 0.02), BMI (*β = -0.11*; 95% CI -0.15 to -0.06; p<0.001) and WC (*β = 0.03*; 95% CI 0.004 to 0.049; p = 0.02), while in the second subgroup (Table 5) SCORE was associated to NC (*β = 0.14*; 95% CI 0.09 to 0.19; p<0.001), BMI (*β = -0.31*; 95% CI -0.39 to -0.23; p<0.001) and WHtR (*β = 10.11*; 95% CI 6.47 to 13.74; p<0.001).

The present study, performed in a population of apparently healthy subjects but with overweight and obesity, with a high prevalence of MHO (48.3%), was aimed at identifying the

**Table 3. Multivariate Poisson linear regression models of SCORE and all anthropometric parameters included in the model (A) Final model applying a backward stepwise method for SCORE on all variables included together in the model (B)-).**

| Parameters [*] | SCORE [§] | | |
|---|---|---|---|
| | β | p-value | C.I. (95%) |
| **A)** | | | |
| NC (cm) | 0.10 | <0.001 | 0.05 to 0.14 |
| BMI (kg/m$^2$) | -0.19 | <0.001 | -0.24 to -0.14 |
| WC (cm) | 0.02 | 0.05 | -0.0004 to 0.0507 |
| WHtR | 6.51 | 0.001 | 2.82 to 10.19 |
| **B)** | | | |
| NC (cm) | 0.15 | <0.001 | 0.12 to 0.18 |
| BMI (kg/m$^2$) | -0.18 | <0.001 | -0.22 to -0.14 |
| WHtR | 7.56 | <0.001 | 5.30 to 9.82 |

[*] Abbreviations: β, coefficient; C.I., Coefficient Interval at 95%; SCORE, Systematic COronary Risk Evaluation; NC, neck circumference; BMI, Body Mass Index; WC, waist circumference; WHtR, waist to height ratio.

anthropometric parameters most clearly associated with cardio-metabolic risk factors such as glucose, lipids, insulin, insulin resistance, vitamin D, hs-CRP, WBC and platelets count, blood pressure, smoking and snoring.

It showed that BMI, WC, WHtR are all very strongly associated with all the parameters studied. The only parameter showing a slight difference was NC; in fact, this parameter was the only one to show a significant, positive association to LDL-cholesterol, but also the only variable not to show a significant association with vitamin D and hs-CRP. Moreover, after a stepwise approach, also NC, followed by BMI and WHtR (or WC in people with Metabolic Syndrome) were associated to the SCORE, the official European cardiovascular disease risk assessment model. In particular, in our population, BMI was inversely associated with SCORE. This inverse association was confirmed also in MHO and Metabolic Syndrome subgroups. This seemingly inconsistent result could be due to the wide age range of our population (with a high proportion of subjects <40 years) and the lack of obesity indices among the parameters

**Table 4. Multivariate Poisson linear regression models of SCORE and all anthropometric parameters included in the model (A). Final model in stepwise method in backward of SCORE on all variables included together in the model (B) in subjects with Metabolic Syndrome (36.9%).**

| Parameters [*] | SCORE [§] | | |
|---|---|---|---|
| | β | p-value | C.I. (95%) |
| **A)** | | | |
| NC (cm) | 0.05 | 0.15 | -0.02 to 0.11 |
| BMI (kg/m$^2$) | -0.12 | 0.002 | -0.19 to -0.04 |
| WC (cm) | 0.03 | 0.15 | -0.01 to 0.06 |
| WHtR | 1.75 | 0.55 | -3.97 to 7.48 |
| **B)** | | | |
| NC (cm) | 0.06 | 0.02 | 0.01 to 0.10 |
| BMI (kg/m$^2$) | -0.11 | <0.001 | -0.15 to -0.06 |
| WC (cm) | 0.03 | 0.02 | 0.004 to 0.049 |

[*] Abbreviations: β, coefficient; C.I., Coefficient Interval at 95%; SCORE, Systematic COronary Risk Evaluation; NC, neck circumference; BMI, Body Mass Index; WC, waist circumference; WHtR, waist to height ratio.

**Table 5. Multivariate Poisson linear regression models of SCORE and all anthropometric parameters included in the model (A). Final model in stepwise method in backward of SCORE on all variables included together in the model (B) in subjects with MHO (48.3%).**

| Parameters * | SCORE $^\S$ | | |
|---|---|---|---|
| | β | p-value | C.I. (95%) |
| A) | | | |
| NC (cm) | 0.09 | 0.03 | 0.01 to 0.17 |
| BMI (kg/m$^2$) | -0.31 | <0.001 | -0.40 to -0.21 |
| WC (cm) | 0.03 | 0.26 | -0.02 to 0.07 |
| WHtR | 8.44 | 0.006 | 2.48 to 14.40 |
| B) | | | |
| NC (cm) | 0.14 | <0.001 | 0.09 to 0.19 |
| BMI (kg/m$^2$) | -0.31 | <0.001 | -0.39 to -0.23 |
| WHtR | 10.11 | <0.001 | 6.47 to 13.74 |

* Abbreviations: β, coefficient; C.I., Coefficient Interval at 95%; SCORE, Systematic COronary Risk Evaluation; NC, neck circumference; BMI, Body Mass Index; WC, waist circumference; WHtR, waist to height ratio.

considered in the SCORE assessment (gender, age, total cholesterol, systolic blood pressure and smoking habit). Moreover, it is well known that BMI is not a good predictor of BFD [7], which is more important than BMI in defining the CVD risk [7]. Furthermore, even though the so-called obesity paradox in CVD has been well described, where CVD patients with obesity have a greater prognosis than their normal weight counterparts do [27], this paradox has not been confirmed when estimating the risk of having CVD [28].In fact, maintaining a healthy weight is one of the main indications in all guidelines for the prevention of CVDs, including those drawn up by the European Society of Cardiology [20]. For this reason, further studies, including an assessment of body composition, are needed to investigate this inverse association found in our population. To the best of our knowledge, the only study (the SOON cohort) which compared anthropometric parameters to identify the most appropriate cardio-metabolic risk marker in subjects with obesity found that NC was the best one [17]. However, firstly this study investigated only women and only patients with severe obesity and, therefore, a community that is not representative of the whole population with overweight and obesity [16]. Secondly, it included patients affected by hypertension and/or type 2 diabetes and/or OSAS, and taking drugs. Thirdly, in the SOON cohort WHtR was not examined [17]. When evaluating our findings that BMI, WC and WHtR showed a similar statistical power in association to cardio-metabolic risk factors, it is important to remember that BMI does not take into account BDF, as opposed to WC or WHtR. Therefore, both BMI and WC (or WHtR) should be measured in subjects with excess body weight. Moreover, although some studies showed that WHtR was more predictive of CVD than BMI and WC [13,14], the present study does not seem to confirm this finding. Thus, WHtR may be a useful further anthropometric parameter to define a patient with overweight or obesity, but does not seem to be superior to WC for this purpose. In addition, it is noteworthy that studies emphasizing the role of WHtR more than BMI and WC were performed in people from Asia and in Caucasian subjects [13,14]. On the other hand, studies in Western populations showed that WC is the best adiposity measure in predicting CVD risk factors [29,30], although a study of a Spanish Mediterranean population continued to support the BMI [31]. Moreover, a systematic review and meta-analysis of Caucasian populations, also assessing WHtR, concluded that WC was more strongly associated with CVD risk factors, and therefore recommended the use of WC in both the clinic and research studies [32]. In addition, NC, that is the anthropometric parameter commonly used in patients

with snoring and suspected OSAS, was shown to have a strong statistical power such as BMI, WC and WHtR, in the stepwise model in association to the SCORE, but not in relationship to each single variable evaluated. Moreover, it was the only parameter not to show a significant correlation with vitamin D and CRP. Even though recent studies, performed in a Chinese population [33] and in young Spanish adults [34], showed that NC has the same power as waist circumference in identifying metabolic disorders and quantifying cardiovascular risk, it has been stated that an increased NC is positively associated with the metabolic syndrome factors; thus the risk of coronary heart disease is likely to increase [35]. However, we agree with Caro et al, who suggested that NC measurement may be an opportunity in clinical practice when it is difficult to measure WC [36]. Interestingly, in this study, NC was the only anthropometric parameter to show a significant correlation with total LDL-cholesterol, and this result is in line with a recent systematic review and meta-analysis [37].

On the basis of all our findings, NC, combined with BMI and WC or WHtR, should be used in clinical practice to quantify the cardio-metabolic risk in individuals with excess body weight. It is notable that several studies have suggested that both BMI and WC should be measured, since the full strength of the association between waist circumference with morbidity and mortality is observed only after adjustment for BMI [38,39]. Indeed, the Consensus by IAS and ICCR et al suggested that the measurement of both BMI and WC should be included when stratifying obesity- related health risk [8].

As regards the strong points of this study, in our opinion, the first is the consistency of the anthropometric data with most of the biomarkers evaluated, an evident sign of the internal validity of the study. Moreover, we examined a large population, involving 1214 subjects, who were not taking any medication that could interfere with anthropometric and biological markers. There are few studies available with all these features. On the other hand, we did not include WHR in this study, since unfortunately this parameter was not available in all the subjects under study. Two limitations of our study are the imbalance between the number of men and women, and the broad age range of the study population. These could be due to the fact that the subjects spontaneously presented to our clinic for reasons of excessive weight, thus representing a selection bias. Moreover, the SCORE, having an age range starting from 40 years, is not entirely applicable to our study population, that includes many young subjects. Another weak point of this study is that it was a cross-sectional investigation, while prospective studies should be carried out to state whether WHtR and NC add information as compared to WC and BMI.

## Conclusion

In conclusion, the present study, performed in a large group of uncomplicated subjects with overweight or obesity, suggests that NC, combined with BMI and WC or WHtR could represent essential tools for use in clinical practice to define the cardio-metabolic risk in individuals with excess body weight. To confirm these findings, the same measures should be used in a prospective study, assessing standardized populations, and with standardized outcomes such as death or cardiovascular disorders, with a precise time-event relationship.

## Author Contributions

**Conceptualization:** Giovanni De Pergola.

**Data curation:** Rossella Donghia, Vito Guerra.

**Formal analysis:** Rossella Donghia, Vito Guerra.

**Investigation:** Roberta Zupo, Fabio Castellana, Isanna Murro, Carmen Di Noia.

**Methodology:** Giovanni De Pergola.

**Project administration:** Rodolfo Sardone.

**Supervision:** Gianluigi Giannelli.

**Validation:** Rodolfo Sardone.

**Visualization:** Luisa Lampignano.

**Writing – original draft:** Luisa Lampignano, Giovanni De Pergola.

**Writing – review & editing:** Rodolfo Sardone.

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
