## [Decision Letter · Decision Letter 0]

21 Jul 2020

PONE-D-20-14485

Cross-Sectional Relationship Among Different Anthropometric Parameters and Cardio-metabolic Risk Factors in a Cohort of Patients with Overweight or Obesity

PLOS ONE

Dear Dr. De Pergola, Dear Giovanni,

Thank you for submitting your manuscript to PLOS ONE. After careful consideration, we feel that it has merit but does not fully meet PLOS ONE’s publication criteria as it currently stands. You'll find below the comments of the reviewers and this Editor. Therefore, we invite you to submit a revised version of the manuscript that addresses the points raised during the review process.

We look forward to receiving your revised manuscript.

Kind regards,

Michele Vacca, M.D., Ph.D.

Academic Editor

PLOS ONE

Journal Requirements:

2. Please address the following:

- Please include additional information regarding the survey or questionnaire used in the study and ensure that you have provided sufficient details that others could replicate the analyses.

For instance, if you developed a questionnaire as part of this study and it is not under a copyright more restrictive than CC-BY, please include a copy, in both the original language and English, as Supporting Information.

- Please consider checking the abstract for errors of grammar.

- Please ensure you have thoroughly discussed all potential limitations of this study within the Discussion section, including any biases introduced during data collection.

5. Your ethics statement must appear in the Methods section of your manuscript. If your ethics statement is written in any section besides the Methods, please move it to the Methods section and delete it from any other section. Please also ensure that your ethics statement is included in your manuscript, as the ethics section of your online submission will not be published alongside your manuscript.

**Editor Comments:**

Dear Authors,

please accept our apologises for the slight delay in the revision process.

The editorial team has extensively reviewed the manuscript and the reviewers and myself see merit in the data; however additional analyses have been suggested (that are largely reasonable) and it has been suggested that you invest more efforts in clearly pointing to the novelty of your data: as the reviewer 2 has pointed there is extensive literature in the filed and, despite PlosOne is more interested to scientific rigour than novelty, better defining what the angle that the authors consider more novel is really important. It is also important to clearly state what are the limitations of the study in the discussion.

Moreover, I think that the reviewers have provided useful suggestions: given the young age of the population and the relatively mild disease, we expect that the number of CVD cases will be really limited so as the manuscript is, it is difficult to dissect the "cardio" from the "metabolic" angle; however, as the reviewer 1 has suggested, CVR can be estimated using Framingham or Progetto Cuore. There will be a proportion of patients falling below the age limits (30 and 35 yo, respectively) for these calculators but those patients can be either excluded from the calculations or be attributed a risk of "0" (assuming the authors clearly state in the methods how missing data are treated). Also, NAFLD is an indipendent CVD risk factor: is liver US available for those subjects? Otherwise the authors can consider to use the Fatty Liver Index or similar risk scores to provide an "estimate".

Please add the beta coefficient in the abstract when discussing the multivariate results. There is only the C.I. and P at the moment. 

I think that most of the suggestions of the reviewers are easily addressable. We thus look forward for receiving a revised version of the manuscript with those addressed.

Best Wishes

Michele Vacca

**Reviewers' comments:**

Reviewer's Responses to Questions

**Comments to the Author**

1. Is the manuscript technically sound, and do the data support the conclusions?

Reviewer #1: Partly

Reviewer #2: Yes

2. Has the statistical analysis been performed appropriately and rigorously? 

Reviewer #1: Yes

Reviewer #2: I Don't Know

3. Have the authors made all data underlying the findings in their manuscript fully available?

Reviewer #1: Yes

Reviewer #2: No

4. Is the manuscript presented in an intelligible fashion and written in standard English?

Reviewer #1: Yes

Reviewer #2: Yes

5. Review Comments to the Author

Reviewer #1: The manuscript by Lampignano and colleagues reports the results of a cross-sectional study investigating the correlation between anthropometric parameters of obesity/adiposity (i.e. neck circumference, BMI, waist circumference, waist-to-height-ratio) with clinical parameters associated with incident risk of cardiovascular events. By studying a large cohort (n=1214) of overweight and obese individuals, the authors have identified significant age- and gender-adjusted associations between the investigated anthropometric parameters and blood pressure, lipid profile, insulin and HOMA index, leukocyte counts and snoring. The authors conclude that “BMI and WC seem to be the essential anthropometric parameters for use in clinical practice to quantify the cardio-metabolic risk in individuals with overweight and obesity” (as per abstract).

The topic of the study has clinical relevance. It investigates anthropometric parameters, which are easy to implement in clinical practice, with the aim of better grading the cardiometabolic risk in obese patients. A clear strength of the study is the recruitment of a large cohort which is very well characterized for cardiometabolic parameters. However, some aspects limit the current submission and could be addressed to improve the manuscript. Namely:

1) The authors conclude that “…BMI and WC seem to be the essential anthropometric parameters for use in clinical practice to quantify the cardio-metabolic risk…”. However, the current study is only observational and exploited pair-wise associations – yet, age- and gender-corrected - between the anthropometric and cardiovascular parameters. How did the authors identify BMI and WC (among the investigated parameters) as the “essential ones”? Analyses and comparison of the strength of associations with the clinical parameters could be performed to support this conclusion.

2) The authors have assessed pairwise correlations with clinical parameters which are associated with incidence of future cardiovascular events. It would be interesting to know whether these anthropometric parameters also associate with the cardiovascular risk of these patients computed using standardized algorithms (e.g. FRS, SCORE, “Progetto Cuore”). Moreover, the individuation of the antropometric parameter with the strong association (thus, potentially more relevant for clinical application) would significantly improve the findings of the authors.

3) Is there collinearity among the tested parameters? If collinearity is not relevant, the authors could also consider enter all the anthropometric parameters into the regression models (also with a stepwise approach) to verify whether some of them could independently associate with the clinical parameters (and/or the estimated cardiovascular risk, see point #2).

4) Which is the percentage of patients fulfilling the diagnostic criteria for Metabolic Syndrome and Metabolically healthy obesity? Would the association still be significant in these two subpopulations?

5) ll.87-88: HOMA-IR and 25-OH-VitD are included among the parameters associated with CV risk. A reference should be included here.

Reviewer #2: The authors present a cross-sectional analysis of four anthropometric indices of body fat distribution and a range of cardiometabolic risk factors amongst over 12 hundred overweight/obese patients. The study aimed to identify the anthropometric parameter(s) most closely associated with cardiometabolic risk, and therefore best predictive of cardiovascular disease. Using a combination of linear and logistic regression models for each anthropometric parameter, the authors demonstrated that BMI, waist circumference, weight-height ratio and neck circumference were positively associated with the majority of cardiometabolic variables, including blood pressure, fasting glucose and insulin/HOMA-IR. A number of other associations were explored. The authors concluded that BMI and WC are the most appropriate anthropometric indices to use in clinical practice to quantify cardiovascular risk.

This is a solid dataset of drug-naïve individuals with overweight or obesity and no documented cardiac or metabolic disease. All individuals have undergone comprehensive anthropometric and biochemical evaluation. The selection and determination of analytes is appropriate. The study is relatively small (n= 1214) for the question being asked, however it is appropriately designed to meet the stated objectives. The authors have used simple linear regression (or logistic regression for snoring/smoking) to model the relationship between anthropometric indices and cardiometabolic risk factors. Confounders such as age and sex were appropriately adjusted for. The conclusions drawn are consistent with the results of these analyses.

Major issues (in order of importance)

1. The use of anthropometric indices of body fat distribution to identify healthy individuals at increased cardiometabolic risk has been widely evaluated in a range of studies, most of them cross-sectional, many significantly larger that this one, and collectively spanning a range of ethnic groups. A subset of these studies are referenced in this paper. Many of these studies have demonstrated that measurement of waist circumference, waist-hip ratio or weight-height ratio affords superior predictive value than BMI alone. I am not therefore convinced that this study adds usefully to the existing literature.

2. Given the question being asked and the clinical context in which anthropometric measurements are used, the use of receiver operating characteristic (ROC) analyses would be appropriate to evaluate the discriminatory power of BMI, WC, WtHR and NC.

3. More information should be presented about the study participants. What was the ethnic composition? What was the age range? Indeed, it is not clearly stated that all participants were adults. What was the BMI range? For many of these parameters, simply reporting mean and SD is insufficient and makes it difficult have confidence in the statistical approach.

4. As the authors already point out, this is a cross sectional study that does not capture cardiovascular outcomes.

Minor issues

1. Graphical representations of the data/regression models would enhance the manuscript.

2. Depending on the BMI distribution of the participants, it would interesting to see if the models hold up at the higher BMI ranges (e.g. BMI >30).

3. The male/female imbalance in this study is unexpected. The authors have not commented on potential reasons for this. There is an opportunity here to repeat the analyses in males vs females and see if any differences arise.

4. There are some sections, particularly in the discussion, in which the wording could be improved to improve clarity.

6. PLOS authors have the option to publish the peer review history of their article (what does this mean?). If published, this will include your full peer review and any attached files.

Reviewer #1: No

Reviewer #2: No

---

## [Author Response · Author response to Decision Letter 0]

30 Aug 2020

Answers to Reviewer #1

Firstly, we would like to thank the reviewer for having carefully reviewed our manuscript and for the useful advice given.

1) The authors conclude that “…BMI and WC seem to be the essential anthropometric parameters for use in clinical practice to quantify the cardio-metabolic risk…”. However, the current study is only observational and exploited pair-wise associations – yet, age- and gender-corrected - between the anthropometric and cardiovascular parameters. How did the authors identify BMI and WC (among the investigated parameters) as the “essential ones”? Analyses and comparison of the strength of associations with the clinical parameters could be performed to support this conclusion.

We totally agree with the reviewer. For this purpose, we extensively modified the text, in the light of the new analysis performed. Please read the next points, in which all the changes made are explained

2) The authors have assessed pairwise correlations with clinical parameters which are associated with incidence of future cardiovascular events. It would be interesting to know whether these anthropometric parameters also associate with the cardiovascular risk of these patients computed using standardized algorithms (e.g. FRS, SCORE, “Progetto Cuore”). Moreover, the individuation of the anthropometric parameter with the strong association (thus, potentially more relevant for clinical application) would significantly improve the findings of the authors.

3) Is there collinearity among the tested parameters? If collinearity is not relevant, the authors could also consider enter all the anthropometric parameters into the regression models (also with a stepwise approach) to verify whether some of them could independently associate with the clinical parameters (and/or the estimated cardiovascular risk, see point #2).

2-3. For this purpose, we estimated cardiovascular risk by the SCORE Risk Charts and we combined all the anthropometric parameters into the regression models with a stepwise approach to verify whether some of them could be independently associated with the SCORE Risk Charts. We used these charts because they are the most commonly used, and validated in European populations. We also decided to add and discuss the new results of this analysis in our manuscript (see Result section lines 193-197 and Discussion section lines 210-211 and 213-219).

4) Which is the percentage of patients fulfilling the diagnostic criteria for Metabolic Syndrome and Metabolically healthy obesity? Would the association still be significant in these two subpopulations?

4. The prevalence of Metabolic Syndrome was 36.9% and prevalence of MHO (obese subject without Metabolic Syndrome) was 48.3%. We have also carried out the analyses in the sub-groups you mentioned. Please find the results shown at the end of this document (Table rev1a and Table rev1b)

5) ll.87-88: HOMA-IR and 25-OH-VitD are included among the parameters associated with CV risk. A reference should be included here.

5. We agree with the reviewer. Two references have been inserted:

- regarding HOMA-IR, in line 86 (17) Ormazabal V, Nair S, Elfeky O, Aguayo C, Salomon C, Zuñiga FA. Association between insulin resistance and the development of cardiovascular disease. Cardiovasc Diabetol. 2018;17(1):122. Published 2018 Aug 31. doi:10.1186/s12933-018-0762-4 

- regarding Vitamin D, in line 87 (18) Muscogiuri G, Annweiler C, Duval G, et al. Vitamin D and cardiovascular disease: From atherosclerosis to myocardial infarction and stroke. Int J Cardiol. 2017;230:577-584. doi:10.1016/j.ijcard.2016.12.053

Table rev1a. Univariate linear regression models of SCORE and anthropometric parameters in patients with Metabolic Syndrome.

Parameters * SCORE § 

 β p-value C.I. (95%) 

Neck (cm) 0.03 0.20 -0.01 to 0.07 

BMI (kg/m²) -0.04 0.001 -0.07 to -0.02 

Waist (cm) -0.005 0.34 -0.01 to 0.005 

Waist to Height Ratio -2.02 0.02 -3.70 to -0.34 

§ Poisson regression Model. 

* Abbreviations: β, coefficient; C.I., Coefficient Interval at 95%; Systematic COronary Risk Evaluation (SCORE); FLI, BMI, Body Mass Index; BMI, Body Mass Index.

Table rev1b. Univariate linear regression models of SCORE and anthropometric parameters in MHO patients 

Parameters * SCORE § 

 β p-value C.I. (95%) 

Neck (cm) 0.06 0.04 0.002 to 0.113 

BMI (kg/m²) -0.14 <0.001 -0.20 to -0.08 

Waist (cm) 0.005 0.53 -0.01 to 0.02 

Waist to Height Ratio 0.26 0.84 -2.32 to 2.84 

§ Poisson regression Model. 

* Abbreviations: β, coefficient; C.I., Coefficient Interval at 95%; Systematic COronary Risk Evaluation (SCORE); BMI, Body Mass Index; BMI, Body Mass Index.

Answers to Reviewer #2

Firstly, we would like to thank the reviewer for having accurately reviewed our paper and for giving useful advices.

Major issues:

1. The use of anthropometric indices of body fat distribution to identify healthy individuals at increased cardiometabolic risk has been widely evaluated in a range of studies, most of them cross-sectional, many significantly larger than this one, and collectively spanning a range of ethnic groups. A subset of these studies are referenced in this paper. Many of these studies have demonstrated that measurement of waist circumference, waist-hip ratio or weight-height ratio affords superior predictive value than BMI alone. I am not therefore convinced that this study adds usefully to the existing literature.

We appreciate the frank comment. However, at variance with most of the other papers, we simultaneously examined 4 anthropometric parameters, and NC had the strongest association with SCORE. In addition, this study was performed in a specific population of people with overweight or obesity, none of which was taking any kind of drug. The sum of these four aspects makes this study quite original.

2. Given the question being asked and the clinical context in which anthropometric measurements are used, the use of receiver operating characteristic (ROC) analyses would be appropriate to evaluate the discriminatory power of BMI, WC, WtHR and NC.

Unfortunately, we cannot proceed with the ROC analysis because of the design of our study, since we do not have a dichotomic variable as outcome. 

3. More information should be presented about the study participants. What was the ethnic composition? What was the age range? Indeed, it is not clearly stated that all participants were adults. What was the BMI range? For many of these parameters, simply reporting mean and SD is insufficient and makes it difficult have confidence in the statistical approach.

We have added ranges to Table 1. Moreover, we have added the ethnic composition in the methods section (line 98, “all Caucasian”) and the range information you requested in the Results section (lines 168-170) and Table 1 “Table 1 shows the general, anthropometric, hormone, metabolic and routine biochemical characteristics of the enrolled population. As reported in the new text, our study population consisted of 31% men, mean age was about 40 years (range 14-70 years) and mean BMI was 34 kg/m2 (range 25-64.4)”

Minor issues:

1. Graphical representations of the data/regression models would enhance the manuscript.

Unfortunately, we cannot display graphic output of the analysis since they are linear regression models.

2. Depending on the BMI distribution of the participants, it would be interesting to see if the models hold up at the higher BMI ranges (e.g. BMI >30).

Below we show the table that explains the models considering only those subjects with a BMI over 30. All the models hold up, except those concerning inflammation parameters. The table (Table rev2) is shown at the end of this document.

3. The male/female imbalance in this study is unexpected. The authors have not commented on potential reasons for this. There is an opportunity here to repeat the analyses in males vs females and see if any differences arise.

We have added some comments on this imbalance in the final part of the discussion section (lines 263-266): “Two limitations of our study are the imbalance between the number of men and women and the broad age range of the study population. These could be due to the fact that the subjects spontaneously presented to our clinic for reasons of excessive weight, thus representing a selection bias”

Moreover, all association models were adjusted for gender. This means that all associations were estimated ceteris paribus of gender.

4. There are some sections, particularly in the discussion, in which the wording could be improved to improve clarity.

Following your advice, we have proceeded to have the manuscript again corrected by our native English speaker collaborator.

Table rev2. Multivariate regression models # between continuous § and categorical ψ variables and anthropometric parameters in patients with BMI>30.

 Neck (cm) BMI (Kg/m2) Waist (cm) Waist to Height Ratio

Parameters * β p-value C.I. (95%) β p-value C.I. (95%) β p-value C.I. (95%) β p-value C.I. (95%)

Continuous Variables § 

DBP (mmHg) 0.35 0.001 0.13 to 0.57 0.13 0.02 0.02 to 0.25 0.10 0.001 0.04 to 0.14 10.18 0.01 2.47 to 17.89

SBP (mmHg) 0.52 0.001 0.21 to 0.82 0.37 <0.001 0.21 to 0.53 0.17 <0.001 0.10 to 0.24 20.60 <0.001 9.53 to 31.64

Total Cholesterol (mg/dL) 0.40 0.32 -0.39 to 1.19 -0.05 0.82 -0.49 to 0.39 -0.001 0.99 -0.19 to 0.19 4.42 0.77 -25.35 to 34.20

Triglyceride (mg/dL) 3.55 <0.001 2.20 to 4.90 2.02 <0.001 1.30 to 2.75 0.85 <0.001 0.54 to 1.17 117.65 <0.001 68.42 to 166.88

FBG (mg/dL) 0.57 <0.001 0.29 to 0.84 0.20 0.01 0.04 to 0.35 0.10 0.003 0.03 to 0.16 11.63 0.02 1.44 to 21.82

HDL (mg/dL) -0.88 <0.001 -1.13 to -0.64 -0.33 <0.001 -0.47 to -0.20 -0.16 <0.001 -0.22 to -0.10 -21.21 <0.001 -30.34 to -12.07

Insulin (mg/dL) 1.83 <0.001 1.46 to 2.21 1.11 <0.001 0.91 to 1.31 0.43 <0.001 0.34 to 0.52 57.55 <0.001 43.66 to 71.44

HOMA-IR 0.47 <0.001 0.37 to 0.57 0.27 <0.001 0.22 to 0.32 0.10 <0.001 0.08 to 0.13 14.00 <0.001 10.41 to 17.58

LDL Cholesterol (mg/dL) 0.64 0.08 -0.08 to 1.37 -0.16 0.44 -0.55 to 0.24 -0.005 0.95 -0.18 to 0.17 0.03 0.99 -26.62 to 26.67

Platelets (103/µL) 1.06 0.17 -0.45 to 2.57 0.82 0.04 0.04 to 1.61 0.35 0.04 0.01 to 0.69 38.12 0.15 -14.44 to 90.69

WBC (103/µL) 0.05 0.02 0.01 to 0.09 0.02 0.10 -0.004 to 0.041 0.01 0.06 -0.0003 to 0.019 0.90 0.23 -0.59 to 2.39

Vitamin D (ng/dL) -0.01 0.94 -0.37 to 0.35 -0.17 0.12 -0.39 to 0.05 -0.06 0.22 -0.16 to 0.04 -3.77 0.62 -18.69 to 11.14

CRP (mg/dL) 0.003 0.67 -0.01 to 0.02 0.006 0.13 -0.002 to 0.014 0.003 0.05 -0.0004 to 0.007 0.49 0.07 -0.03 to 1.01

 Neck (cm) BMI (Kg/m2) Waist (cm) Waist to Height Ratio

Parameters * OR p-value C.I. (95%) OR p-value C.I. (95%) OR p-value C.I. (95%) OR p-value C.I. (95%)

Dichotomic Variables ψ 

Snoring 1.17 <0.001 1.11 to 1.25 1.10 <0.001 1.06 to 1.13 1.04 <0.001 1.03 to 1.05 341.58 <0.001 45.94 to 2539.82

Smoking 1.03 0.29 0.97 to 1.09 1.01 0.45 0.98 to 1.04 1.00 0.38 0.99 to 1.02 2.50 0.39 0.31 to 19.93

§ Multivariate Linear Regression Model; ψ Multivariate Logistic Regression Model.

# Adjusted for Age and Gender.

* Abbreviations: β, coefficient; C.I., Coefficient Interval at 95%; OR, Odds Ratios; BMI, Body Mass Index; IMT, Intima-media thickness; DBP, Diastolic Blood Pressure; SBP, Systolic Blood Pressure; FBG, Fasting Blood Glucose; HDL, Hight 

 Density Lipoprotein; HOMA-IR, Homeostasis Model Assessment-Insulin Resistance; LDL, Low Density Lipoprotein; WBC, White Blood Cell; CRP, C-Reactive Protein.

---

## [Decision Letter · Decision Letter 1]

5 Oct 2020

PONE-D-20-14485R1

Cross-Sectional Relationship Among Different Anthropometric Parameters and Cardio-metabolic Risk Factors in a Cohort of Patients with Overweight or Obesity

PLOS ONE

Dear Prof. De Pergola, Dear Giovanni,

Thank you for submitting your manuscript to PLOS ONE. Please accept my apologise for the delay but, to ensure that your manuscript was seen by the original reviewers, we had to wait for one of them that asked for an extension of the deadline.

I am happy to confirm that the reviewers have found the manuscript largely improved. However, the Reviewer 1 has still a couple of minor (but important) queries that should not take long for you to address before the manuscript can be formally accepted. Therefore, we invite you to submit a revised version of the manuscript that addresses these comments: we will then proceed with an expedite acceptance at the next round.

We look forward to receiving your revised manuscript.

Kind regards,

Michele Vacca, M.D., Ph.D.

Academic Editor

PLOS ONE

Reviewers' comments:

Reviewer's Responses to Questions

**Comments to the Author**

1. If the authors have adequately addressed your comments raised in a previous round of review and you feel that this manuscript is now acceptable for publication, you may indicate that here to bypass the “Comments to the Author” section, enter your conflict of interest statement in the “Confidential to Editor” section, and submit your "Accept" recommendation.

Reviewer #1: (No Response)

Reviewer #2: (No Response)

2. Is the manuscript technically sound, and do the data support the conclusions?

Reviewer #1: (No Response)

Reviewer #2: Yes

3. Has the statistical analysis been performed appropriately and rigorously? 

Reviewer #1: (No Response)

Reviewer #2: Yes

4. Have the authors made all data underlying the findings in their manuscript fully available?

Reviewer #1: (No Response)

Reviewer #2: Yes

5. Is the manuscript presented in an intelligible fashion and written in standard English?

Reviewer #1: Yes

Reviewer #2: Yes

6. Review Comments to the Author

Reviewer #1: The manuscript by Lampignano and colleagues reports the results of a cross-sectional study investigating the correlation between anthropometric parameters of obesity/adiposity (i.e. neck circumference, BMI, waist circumference, waist-to-height-ratio) with clinical parameters associated with incident risk of cardiovascular events.

The revised manuscript is more complete and includes new analyses for evaluating the association of anthropometric parameters with the SCORE risk chart categories of cardiovascular risk. The overall manuscript is improved however there are some aspects that would require further clarification:

1) The authors have employed Poisson regression to regress the score risk category with the anthropometric parameters (new table 3). The authors conclude “… when associated to the SCORE, the official European cardiovascular disease risk assessment model, NC had the strongest association (beta=0.15; 95% CI 0.12 to 0.18; p<0.001), followed by BMI (beta=-0.18; 95% CI -0.22 to 0.14; p<0.001) and WHtR (beta=7.56; 95% CI 5.30 to 9.82; p<0.001)…” (ll.221-223). It is unclear why the value showing the lowest absolute beta-coefficient is deemed as the one with the strongest association. Could the authors verify and clarify?

2) Another interesting aspect raising from the Poisson regression is the different direction of the association of the SCORE risk with NC (beta= +0.15) and BMI (beta=-0.18). This result is interesting, as BMI (and, thus, obesity) is usually associated with higher cardiovascular risk, while these results apparently suggest that lower BMI are associated with higher SCORE risk categories. Could the authors comment on these aspects?

3) The data on MHO and Metabolic Syndrome are not presented in the manuscript and the presented tables are apparently univariate associations. The authors could consider perform a multivariate regression (as in table 3) for the subgroups and perhaps include a short description in the manuscript. These data may be interesting for the readers

Reviewer #2: The authors have made some important changes to this manuscript in response to comments, the most notable of which was incorporation of a the previously-validated Systemic Coronary Risk Evaluation score, which adds another interesting (and novel) element to their work. In the original version, BMI and waist circumference were concluded to be the key anthropometric parameters for use in clinical practice, whereas the revised conclusion promotes neck circumference "in combination" with the other three parameters. Whilst the conclusion is modest, the analyses are carefully done and add usefully to the existing body of work seeking to identify robust anthropometric predictors of cardiovascular outcome.

All comments have been appropriately addressed, in particular the provision of useful information and summary data about the study population.

7. PLOS authors have the option to publish the peer review history of their article (what does this mean?). If published, this will include your full peer review and any attached files.

Reviewer #1: No

Reviewer #2: No

---

## [Author Response · Author response to Decision Letter 1]

16 Oct 2020

Firstly, we would like to thank the reviewers for your suggestions and for the precious time spent reviewing our manuscript. We believe that we have improved our paper thanks to your indications. 

In particular, here are the responses to the comments of the first reviewer.

1) The authors have employed Poisson regression to regress the score risk category with the anthropometric parameters (new table 3). The authors conclude “… when associated to the SCORE, the official European cardiovascular disease risk assessment model, NC had the strongest association (beta=0.15; 95% CI 0.12 to 0.18; p<0.001), followed by BMI (beta=-0.18; 95% CI -0.22 to 0.14; p<0.001) and WHtR (beta=7.56; 95% CI 5.30 to 9.82; p<0.001)…” (ll.221-223). It is unclear why the value showing the lowest absolute beta-coefficient is deemed as the one with the strongest association. Could the authors verify and clarify?

The reviewer is right, we made a mistake in defining the NC as the one with the strongest association. Accordingly, we modified the relative parts of the text in the abstract, results and discussions sections where the concept was expressed.

2) Another interesting aspect arising from the Poisson regression is the different direction of the association of the SCORE risk with NC (beta= +0.15) and BMI (beta=-0.18). This result is interesting, as BMI (and, thus, obesity) is usually associated with higher cardiovascular risk, while these results apparently suggest that lower BMI are associated with higher SCORE risk categories. Could the authors comment on these aspects?

We have expanded the discussion about this finding. In fact, on page 13, lines 241-256 we stated “Moreover, after a stepwise approach, also NC, followed by BMI and WHtR (or WC in people with Metabolic Syndrome) were associated to the SCORE, the official European cardiovascular disease risk assessment model. In particular, in our population, BMI was inversely associated with SCORE. This inverse association was confirmed also in MHO and Metabolic Syndrome subgroups. This seemingly inconsistent result could be due to the wide age range of our population (with a high proportion of subjects <40 years) and the lack of obesity indices among the parameters considered in the SCORE assessment (gender, age, total cholesterol, systolic blood pressure and smoking habit). Moreover, it is well known that BMI is not a good predictor of BFD (7), which is more important than BMI in defining the CVD risk (7). Furthermore, even though the so-called obesity paradox in CVD has been well described, where CVD patients with obesity have a greater prognosis than their normal weight counterparts do (27) , this paradox has not been confirmed when estimating the risk of having CVD(28).In fact, maintaining a healthy weight is one of the main indications in all guidelines for the prevention of CVDs, including those drawn up by the European Society of Cardiology. (20) For this reason, further studies, including an assessment of body composition, are needed to investigate this inverse association found in our population”.

3) The data on MHO and Metabolic Syndrome are not presented in the manuscript and the presented tables are apparently univariate associations. The authors could consider performing a multivariate regression (as in table 3) for the subgroups and perhaps include a short description in the manuscript. These data may be interesting for the readers

As suggested by the reviewer, we added the required information and analysis in the new text. Furthermore we introduced the concept of MHO in the introduction (page 2, lines 48-53). “Several studies showed that a subgroup of subjects with obesity may be at significantly lower risk than usually estimated from obesity-related CVDs (6). This subset has been described as Metabolically healthy obesity (MHO) (6). Compared to patients with metabolically unhealthy obesity, individuals with MHO are distinguished by lower liver and visceral fat but higher subcutaneous leg fat content, higher cardiorespiratory fitness and physical activity, insulin sensitivity, lower levels of inflammatory markers, and normal adipose tissue function (6).”

---

## [Decision Letter · Decision Letter 2]

22 Oct 2020

Cross-Sectional Relationship Among Different Anthropometric Parameters and Cardio-metabolic Risk Factors in a Cohort of Patients with Overweight or Obesity

PONE-D-20-14485R2

Dear Prof. De Pergola, Dear Giovanni,

We’re pleased to inform you that your manuscript has been judged scientifically suitable for publication and will be formally accepted for publication once it meets all outstanding technical requirements. Well Done!

Kind regards,

Michele Vacca, M.D., Ph.D.

Academic Editor

PLOS ONE

Reviewers' comments:

Reviewer's Responses to Questions

**Comments to the Author**

1. If the authors have adequately addressed your comments raised in a previous round of review and you feel that this manuscript is now acceptable for publication, you may indicate that here to bypass the “Comments to the Author” section, enter your conflict of interest statement in the “Confidential to Editor” section, and submit your "Accept" recommendation.

Reviewer #1: (No Response)

2. Is the manuscript technically sound, and do the data support the conclusions?

Reviewer #1: Yes

3. Has the statistical analysis been performed appropriately and rigorously? 

Reviewer #1: Yes

4. Have the authors made all data underlying the findings in their manuscript fully available?

Reviewer #1: (No Response)

5. Is the manuscript presented in an intelligible fashion and written in standard English?

Reviewer #1: (No Response)

6. Review Comments to the Author

Reviewer #1: The authors have satisfactorily addressed my comments. The revised manuscript is sensibly improved and better conveys the findings of the study.

7. PLOS authors have the option to publish the peer review history of their article (what does this mean?). If published, this will include your full peer review and any attached files.

Reviewer #1: No

---

## [Editor Report · Acceptance letter]

26 Oct 2020

PONE-D-20-14485R2 

Cross-Sectional Relationship Among Different Anthropometric Parameters and Cardio-metabolic Risk Factors in a Cohort of Patients with Overweight or Obesity 

Dear Dr. De Pergola:

I'm pleased to inform you that your manuscript has been deemed suitable for publication in PLOS ONE. Congratulations! Your manuscript is now with our production department. 

Kind regards, 

on behalf of

Dr. Michele Vacca 

Academic Editor

PLOS ONE